# Data-Rate Constrained Observers of Nonlinear Systems

**DOI:** 10.3390/e21030282

**Published:** 2019-03-14

**Authors:** Quentin Voortman, Alexander Yu. Pogromsky, Alexey S. Matveev, Henk Nijmeijer

**Affiliations:** 1Department of Mechanical Engineering, Eindhoven University of Technology, 5612 AZ Eindhoven, The Netherlands; 2Department of Control Systems and Robotics, Saint Petersburg National Research University of Information Technologies, Mechanics and Optics (ITMO), 197101 Saint Petersburg, Russia; 3Department of Mathematics and Mechanics, Saint Petersburg State University, 198504 Saint Petersburg, Russia

**Keywords:** nonlinear systems, observers, data-rate constraints

## Abstract

In this paper, the design of a data-rate constrained observer for a dynamical system is presented. This observer is designed to function both in discrete time and continuous time. The system is connected to a remote location via a communication channel which can transmit limited amounts of data per unit of time. The objective of the observer is to provide estimates of the state at the remote location through messages that are sent via the channel. The observer is designed such that it is robust toward losses in the communication channel. Upper bounds on the required communication rate to implement the observer are provided in terms of the upper box dimension of the state space and an upper bound on the largest singular value of the system’s Jacobian. Results that provide an analytical bound on the required minimum communication rate are then presented. These bounds are obtained by using the Lyapunov dimension of the dynamical system rather than the upper box dimension in the rate. The observer is tested through simulations for the Lozi map and the Lorenz system. For the Lozi map, the Lyapunov dimension is computed. For both systems, the theoretical bounds on the communication rate are compared to the simulated rates.

## 1. Introduction

Ever since the widespread usage of wireless technologies, there has been a focus on data-rate problems for dynamical systems. These problems arise when networked technologies are employed in configurations where sensors, actuators, and controllers are placed at locations that are remote from one another. Extra complications arise from uncertainties in the system’s parameters, initial conditions, sensor measurements, communication channels, and dynamics, such as in the form of exogenous disturbances. These necessitate communication strategies that are efficient in terms of data rate and robust against all kinds of uncertainty. In this paper, the focus will be placed on uncertainties in the initial conditions, and also on issues of losses in the communication channel.

Up until now, the main focus in the relevant control-oriented literature was on state estimation and stabilization problems. In the early 2000’s, most of the research dealt with linear systems, for which many results have been obtained (see [1,2,3,4] for extended surveys).

Early results on nonlinear systems (exemplified by [5,6]) typically assumed special properties of the considered systems, and were also based on these properties. Proper adapting and extending techniques, originally developed for linear plants, opened the door for handling generic nonlinear systems and permitted the establishment of lower data-rate bounds sufficient for the observability and stabilizability of such systems; see, e.g., [7]. Another research trend was concerned with intrinsic characteristics of nonlinear systems that provide a somewhat exhaustive description of the bit-rate of information transmission under which a certain dynamic property (such as stability, invariance, observability) can be achieved. As a result, there appeared a whole series of extensions and modifications of classical topological entropy [8], such as feedback topological entropy [9], invariance entropy [10,11], topological entropy with regard to dynamic uncertainties [12] and to uncertainties in the initial conditions [13], estimation entropy [14], and others (see, e.g., [15,16,17,18]). Finally, some papers such as [19] have relied on passivity-based methods to provide bit-rate bounds.

One of the objectives of the paper is to use non-Euclidean concepts of the set dimension as an alternative to the aforementioned notions of entropy. Among the non-Euclidean concepts of set dimensions, the best known is, maybe, the Hausdorf dimension [20]. Another related characteristic is the upper box dimension [21], which is sometimes referred to as the limit capacity [22]. These two dimensions—the entropy and Lyapunov exponent—were proven to be closely related to one another in [23,24,25]. Both dimensions are based on covering a set with balls of infinitesimally small size (a technique which is very similar to the idea of partitioning the state-space and using symbolic dynamics to describe the dynamical system—see, e.g., [26]); both can assume non-integer values, and both may be smaller than the dimension of the hosting Euclidean space. These concepts have been much inspired by studies of fractals and research on chaotic attractors of dynamical systems. The latter is a primary incentive for our interest in these dimensions, which may be a non-integer for chaotic attractors and provide a somewhat deeper insight into the issue of their dimensionality than the ordinary Euclidean dimension.

Unfortunately, there are still no general analytical techniques for computing the two aforementioned dimensions for chaotic attractors. Numerical methods remain the main tool used by scientists and engineers to estimate these dimensions [27]. In this paper, an alternative to this numerical approach is developed. The alternative is based on the so-called Lyapunov dimension, in which the upper bounds are the above two dimensions [28]. Moreover, its advantage resides in the fact that the Lyapunov dimension can be computed analytically by using the second Lyapunov method [29,30], which leads to analytical upper bounds. By following this alternative, we obtain a fully analytical lower bound on the communication data-rate under which reliable estimation of the system’s state becomes feasible. When doing so, we consider a generic nonlinear system and focus attention on its behavior within a given invariant set, which may be a chaotic attractor, for example.

Apart from this bound, the design of a particular observer that ensures such an observability is also presented. The observer is composed of a sampler, a quantizer, a data-rate constrained channel, and a decoder. All components interact in order to build estimates of the state at a remote location in real time. The observer can ensure arbitrarily high precision of estimation with a communication rate that remains below the channel capacity. Moreover, the proposed observer is robust against delays and losses in the communication channel, which is a valuable property for applications where delays and losses are a common occurrence. This robustness is achieved without any feedback in the communication channel, which is atypical for most data-rate constrained observers in the current literature [31,32,33,34] and constitutes the novel contribution of the paper.

This paper is both a generalization and an extension of [35,36]. We provide a unified solution for both continuous and discrete-time systems. In addition to providing proofs for all the results that were presented in the two aforementioned conference papers, the problem statement is extended to also include delays in the communication channel.

In Section 2, we define the types of systems to be observed, as well as the observer notations, and provide a definition for observability with data-rate constraints. Section 3 introduces the proposed observer. In Section 4, preliminary criteria for observability of the plant are offered, which are converted into a fully analytical form in Section 5. Section 6 illustrates the general theory via handling two examples: the Lozi map and the Lorenz system. For both systems, the necessary data-rates are computed and simulations that confirm the theoretical results are provided.

## 2. Problem Statement

In this section, we introduce the problem statement. The general setting is that of a dynamical system and two peers connected together via a communication channel. Both peers have full knowledge about the dynamics of the system. Meanwhile, only one of them has direct access to the system’s current state and fully measures it. The task is to provide estimates of the state to the other peer by sending messages through the communication channel. This channel is discrete (i.e., the variety of transmittable messages is finite) and has delays, losses, and limited data-rate. The effects due to data-rate constraints and delays are explicitly modeled in this section, whereas the issue of message losses is discussed separately in Remark 1. Two types of delays will be considered in turn. A *processing delay* is also incurred, since the channel can transmit only a given and finite number of bits *c* per unit time, and so a *B*-bits message can wholly arrive at the receiving end of the channel not earlier than B/c time units after the transmission of this message is commenced. A *transmission delay* is caused by holding up the progress in the ideal routine of bits transfer, which may occur as a result of, e.g., resolving competition with third parties for shared resources of the communication medium or network.

In order to solve the stated problem, we will develop a particular type of observer. In this section, we will only introduce notations concerned with the observer. Operation of its components will be described in the next section.

### 2.1. Observed Dynamical System

We consider a dynamical system ${\left\{\varphi^{t}\right\}}_{t\in T}$ on an open set $S\subset R^{n}$, paying special attention to a certain subset $S_{0}\subset S$. Here,
*T* is the set of time periods, which is either $Z^{+}$ or $R^{+}$;$\varphi^{t}:S\to S$ is the evolution function that gives the system state $x\left(t\right)=\varphi^{t}\left(x_{0}\right)$ at time t∈T, provided that the initial state is $x_{0}$;$S_{0}$ is the focus of our interest in the system.

Specifically, we are interested only in trajectories that start in $S_{0}$ and remain there afterwards:(1)$$x\left(t\right)\in S_{0}\;\forall t\in T.$$

**Assumption** **1.**
*The dynamical system at hands is time-invariant: $\varphi^{t}\circ \varphi^{s}=\varphi^{t+s}\;\forall t,s\in T$.*


**Assumption** **2.**
*The set $S_{0}$ is a bounded forward invariant $\varphi^{t}\left(S_{0}\right)\subset S_{0}\;\forall t\in T$, its closure ${\bar{S}}_{0}$ lies in S.*


Our main interest is in systems for which complex and possibly chaotic long-horizon dynamics arise from rather regular short-horizon behavior. The last feature is partly substantiated by the following.

**Assumption** **3.**
*For any t∈T, the evolution function $\varphi^{t}:S\to S$ is continuously differentiable.*


Depending on the “time-set” option, two types of dynamical systems will be considered.

(1) Discrete time systems: $T=Z^{+}$, and the system evolves as follows:$$\begin{array}{c} x\left(t+1\right)=\psi \left(x\left(t\right)\right)\;t\in Z^{+}, \end{array}$$
where ψ:S→S is a given mapping. In this case,
$$\varphi^{t}\left(\cdot \right):=\underset{t\;\mathrm{times}}{\underset{︸}{\psi (\cdots \psi (\cdot ))}}\;\mathrm{and}\;\psi =\varphi^{1}.$$

Assumption 1 holds, and Assumption 3 is met if, and only if ψ is continuously differentiable.

(2) Continuous time systems: $T=R^{+}$ and the evolution of the system is described by an ordinary differential equation (ODE):(2)$$\begin{array}{c} \dot{x}\left(t\right)=f\left(x\left(t\right)\right)\;t\in R^{+}, \end{array}$$
where $f:S\to R^{n}$ is a continuously differentiable vector field. So for any $x_{0}\in S$, the solution $x(t,x_{0})$ of the Cauchy problem $x\left(0\right)=x_{0}$ for the ODE (Equation 2) exists, and is unique; it can be extended to the right on the maximal interval $\left[0,T\left(x_{0}\right)\right)$. However, Equation (Equation 2) not ineluctably defines a dynamical system on *S*, since, not necessarily, $T(x_{0})=\infty$ for all $x_{0}\in S$. Insofar as the right-hand side of Equation (Equation 2) is not defined outside *S*, this extendability $T(x_{0})=\infty$ means, in particular, that the solution $x\left(t,x_{0}\right),x_{0}\in S$ never attempts to leave the set *S*; i.e., the set *S* is forward invariant. So when dealing with ODE, we always assume that all its solutions that start in *S* at t=0 can be extended on [0,∞) while remaining in *S*.

The following proposition can be proved by retracing the arguments from Section 2.2 in [37].

**Proposition** **1.***Whenever the vector field $f:S\to R^{n}$ is smooth (i.e., continuously differentiable) and the ODE* (Equation 2) *has the just-stated extendability and invariance properties, this ODE gives rise to a dynamical system ${\left\{\varphi^{t}\right\}}_{t\in R^{+}}$ on S ($\varphi^{t}\left(x_{0}\right):=x\left(t,x_{0}\right)$), which satisfies Assumptions* 1 *and* 3*. Moreover, $\varphi^{t}\left(x\right)$ and its first derivatives, with respect to t and x, are continuous functions of t and x.*

### 2.2. Architecture of the Observer, Notations, and General Traits of the Communication Channel

We assumed that the current state x(t) was observed in full at a certain *measurement site* but is needed at time *t* at a remote location, where data can be communicated only via a discrete channel. The channel is discrete in the sense that first, it is constrained to carry messages that are drawn from a finite set, and second, the messages can be communicated only one at a time and, while the channel is busy transmitting a previous message, it is closed for the next transmission.

The purpose of the observer is to arrange and manage transmissions across the channel and to finally build, at time *t* and at the remote location, an estimate $\hat{x}\left(t\right)$ of the current state x(t) with a pre-specified exactness. The formal definition of the last notion is as follows.

**Definition** **1.**
*A number ϵ>0 is called an “exactness of observation”, and if there exists ${\bar{t}}_{0}<\infty$ such that*
$$\begin{array}{c} ∥x\left(t\right)-\hat{x}\left(t\right)∥\leq \epsilon ,\;\forall t\in T:t\geq {\bar{t}}_{0}. \end{array}$$


As is illustrated in Figure 1, an *observer*
O is defined as a composition consisting of a *sampler*S, *quantizer*Q, and *decoder*D; the sampler and quantizer together form a *coder*C:$$O\;\mathrm{is}\;\mathrm{composed}\;\mathrm{of}\;\underset{C}{\underset{︸}{S\;\mathrm{and}\;Q}}\;\mathrm{and}\;D.$$
The sampler and quantizer are built at the measurement site and have access to the dynamics $\left\{\varphi^{t}\right\}$ of the system, the set $S_{0}$, the current state x(t), and the desired exactness of observation ϵ.The decoder is built at the remote site L and has access to the system dynamics $\left\{\varphi^{t}\right\}$, the set $S_{0}$, the desired exactness of observation ϵ, and the messages transmitted across the channel.

The roles and structures of the observer components are as follows.

The Sampler generates the (*sampling*) instants $s_{j}\in T$ (where at every one of these instants $t=s_{j}$, transmission of another message $e(s_{j})$ is initiated):(3)$$\begin{array}{cc} s_{j+1}=S\left({\left\{\varphi^{t}\right\}}_{t\in T},S_{0},x\left(s_{j}\right),s_{j},\epsilon \right)>s_{j}, & s_{0}=0,\;\;\;\;\;\;\;j\in Z^{+}. \end{array}$$

Also, the sampler builds a finite *alphabet*
$A_{j}$ from which the message $e(s_{j})$ should be drawn at time $s_{j}$ for subsequently communicating across the channel:(4)$$\begin{array}{cc} A_{j}=A\left({\left\{\varphi^{t}\right\}}_{t\in T},S_{0},x\left(s_{j}\right),s_{j},\epsilon \right), & \;\;\;\;\;\;\;j\in Z^{+}. \end{array}$$

The alphabet is thus permitted to depend on $s_{j}$.

The Quantizer forms the message $e\left(s_{j}\right)\in A_{j}$ to be dispatched
(5)$$\begin{array}{cc} e\left(s_{j}\right)=Q\left({\left\{\varphi^{t}\right\}}_{t\in T},S_{0},x\left(s_{j}\right),s_{j},\epsilon \right), & \forall j\in Z^{+}. \end{array}$$

The Decoder generates state estimates based on the previously received messages:(6)$$\begin{array}{c} \hat{x}\left(t\right)=D\left({\left\{\varphi^{\tau }\right\}}_{\tau \in T},S_{0},{\left\{\left(e\left(s_{j}\right),{\bar{s}}_{j}\right)\right\}}_{j\in J(t)},\epsilon \right), \end{array}$$
where ${\bar{s}}_{j}$ is the time when the message $e(s_{j})$ arrives at the remote site L and $J\left(t\right):=\left\{j:{\bar{s}}_{j}\leq t\right\}$. If no message has arrived yet, J(t)=∅ and the meaningless ${\left\{\left(e\left(s_{j}\right),{\bar{s}}_{j}\right)\right\}}_{j\in \emptyset }$ is replaced by an arbitrarily pre-specified symbol, e.g., $0\in Z^{+}$. The observer has to fit the constraints and capabilities of the channel, which are as follows.
(c.1)The channel correctly transfers any message $e\left(s_{j}\right)\in A_{j}$ to the receiving end provided that the message processing time $\tau_{j}^{\mathrm{pr}}$ and the size of the message are in balance:
(7)$$\log_{2}\mathrm{card}\left(A_{j}\right)\leq b\left(\tau_{j}^{\mathrm{pr}}\right).$$Here, b(τ) is a channel-dependent function that gives the number of bits processable by the channel during any time period of length τ.(c.2)As the processing time increases to infinity, the average number of bits transmittable per unit of time stabilizes and converges to a certain value $c\in R^{+}$, called the (bit-rate) channel *capacity*:
(8)$$\begin{array}{c} \exists c:=\lim_{\tau \to \infty }\frac{b(\tau )}{\tau }. \end{array}$$(c.3)The channel is closed for the next message until all bits of the current message $e(s_{j})$ have been processed, but is open afterwards.(c.4)On its way to the destination point L, any message $e(s_{j})$ incurs a transmission delay $\tau_{j}^{\mathrm{tr}}$:
(9)$${\bar{s}}_{j}=s_{j}+\tau_{j}^{\mathrm{pr}}+\tau_{j}^{\mathrm{tr}},$$
where ${\bar{s}}_{j}$ is the time when the whole of the message $e(s_{j})$ arrives at L, and the processing time $\tau_{j}^{\mathrm{pr}}$ plays the role of a processing delay here.(c.5)The transmission delays are upper-bounded: $\tau_{j}^{\mathrm{tr}}\leq \tau_{+}^{\mathrm{tr}}<\infty$.

To correctly transmit messages, the sampler should balance the chosen alphabet and the message processing time $\tau_{j}^{\mathrm{pr}}$ in accordance with Equation (Equation 7), and respect the requirement: $s_{j+1}\geq s_{j}+\tau_{j}^{\mathrm{pr}}.$

### 2.3. Observability via Channels with Limited Bit-Rate Capacity

The objective of the observer is to guarantee observability, as defined in the following definition.

**Definition** **2.***A system ${\left\{\varphi^{t}\right\}}_{t\in T}$ is said to be observable on the set $S_{0}$ via a communication channel if, for any ϵ>0, there exists an observer Equations* (Equation 3)–(Equation 6) *that operates via this channel and ensures the requested exactness of observation ϵ for any trajectory satisfying Equation* (Equation 1)*.*

Observability is classically defined as a property of the system itself. However, in the current context, finite data rate makes observability critically dependable on the employed communication channel. So, by following [18,35,36,38], observability is introduced as a property of the pair “system + channel”. In Definition 2, the reference to existence of an observer in fact conveys the idea of most effectively utilizing the properties of the system and the potentialities of the channel, where if their clever use may result in a reliable and exact state estimate at the receiving end of the channel, the pair is sealed with the stamp, “observable”.

## 3. Design of the Proposed Observer

Since we are interested only in trajectories satisfying Equation (Equation 1), our discussion of the observer design is confined to the case where $x\left(s_{j}\right)\in S_{0}\;\forall j$ in Equations (Equation 3)–(Equation 5).

We will introduce an observer that is determined by the following four entities:(e.1)$s_{+}\in T$—The period $s_{j+1}=s_{j}+s_{+}$ between consecutive dispatches of messages via the channel;(e.2)*P*—A symmetric and positive definite n×n-matrix;(e.3)$\delta (\epsilon ,s_{+})>0$—A function of ϵ>0 and $s_{+}$ for which
(10)$${∥\hat{x}-x∥}_{P}\leq \delta \left(\epsilon ,s_{+}\right)\;\mathrm{and}\;\hat{x},x\in S_{0}\Rightarrow {∥\varphi^{t}\left(\hat{x}\right)-\varphi^{t}\left(x\right)∥}_{P}\leq \epsilon \;\forall t\in T,t\leq s_{+}.$$(e.4)${\left\{B_{\delta }^{P}\left(q_{k}\right)\right\}}_{k=1}^{K}$—A finite covering of the compact set ${\bar{S}}_{0}$ with $K=K(\epsilon ,s_{+})$ balls (with respect to the norm ${∥\cdot ∥}_{P}$) centered in $q_{i}\in S_{0}$ and with a radius of $\delta =\delta (\epsilon ,s_{+})$ each.

Here, the centers $q_{i}$ may also depend on $\epsilon ,s_{+}$.

**Lemma** **1.***Let Assumptions 2 and 3 hold, and for any τ∈T, the derivatives of $\varphi^{t}$ are bounded over t∈T,t≤τ and x from some τ-dependent neighborhood of $S_{0}$. Then, a function $\delta (\epsilon ,s_{+})>0$ with the property* (Equation 10) *exists.*

The proof of this lemma simply follows from the continuous differentiability of $\varphi^{t}$ and the boundedness of of the derivates.

The proposed observer operates as follows.

**Procedure 1.** (Observer)
*(o.1)* *The sampler S (Equations* (Equation 3) *and* (Equation 4)*) carries out the following actions:*
$$\begin{array}{ccc} s_{j+1}= & S\left({\left\{\varphi^{t}\right\}}_{t\in T},x\left(s_{j}\right),s_{j},\epsilon \right):=s_{j}+s_{+}, & s_{0}:=0.\\ & A\left({\left\{\varphi^{t}\right\}}_{t\in T},x\left(s_{j}\right),s_{j},\epsilon \right):=\left\{1,\ldots ,K\right\}, & \end{array}$$
*i.e., the alphabet substantiates the numbering of the balls from (e.4).**(o.2)* 
*The quantizer Q finds an element $B_{\delta }^{P}\left(q_{k}\right)$ of the covering from (e.4) that contains $x\left(s_{j+1}\right)=\varphi^{s_{+}}\left(x\left(s_{j}\right)\right)\in S_{0}$ and sends its index k over the channel:*
$$\begin{array}{c} Q({\left\{\varphi^{t}\right\}}_{t\in T},x\left(s_{j}\right),s_{j},\epsilon )=k, \end{array}$$
*(o.3)* 
*The decoder D performs the following operations at time t∈T:*
-*Extracts the index k from the last message received at a time $\theta \leq s_{i}$, where $i:=\lfloor t/\left(s_{+}\right)\rfloor$ (If no message has been received yet, k is assigned an arbitrarily pre-specified value, e.g.,* 1*.).*-
*By using the centers from (e.4), forms the current state estimate*
(11)$$\hat{x}\left(t\right):=\varphi^{\left(t-s_{i}\right)}\left(q_{k}\right).$$




Several comments on this observer are as follows:In (o.2), we do not address the case $x\left(s_{j+1}\right)\notin S_{0}$ due to the reason stated at the onset of the section.The proposed design assumes that both the coder and decoder have access to $s_{+}$ from (e.1) and the covering from (e.4).The observer uses a fixed alphabet {1,…,K}, which is shared by the coder and the decoder.The quantizer sends data about the estimate $q_{k}$ of not the current $x(s_{j})$, but the forward-time state $x(s_{j+1})$, which is computed from the measured $x(s_{j})$ by using the known transition map $\varphi^{s_{+}}\left(\cdot \right)$.The idea behind this relies on the expectation that these data will be received prior to $s_{j+1}$ and put in use at due time, $t=s_{j+1}$. Then, the exactness of estimation will be δ at this time.These data are also used to estimate the state on the subsequent time interval $\left\{t\in T:s_{j+1}\leq t<s_{j+2}\right\}$ via applying the matching transition map to the just-discussed estimate at time $t=s_{j+1}$. By Equation (Equation 10), this guarantees the exactness of estimation $∥x\left(t\right)-\hat{x}{\left(t\right)∥}_{P}\leq \epsilon$ on this interval.

In order for the proposed observer to be able to *operate correctly* via a given communication channel, the message $e(s_{j})$ initiated at time $s_{j}$ should be fully processed and received prior to $s_{j+1}$. (This, in particular, implies that the messages arrive in order: ${\bar{s}}_{j+1}>{\bar{s}}_{j}$.) Due to (c.1), (c.4), and (c.5), correct operation occurs whenever there exists a solution $\tau^{pr}\in T$ to the following two inequalities:(12)$$\log_{2}K\left(\epsilon ,s_{+}\right)\leq b\left(\tau^{\mathrm{pr}}\right),\;s_{+}\geq \tau^{\mathrm{pr}}+\tau_{+}^{\mathrm{tr}}.$$

We recall that $\tau_{+}^{\mathrm{tr}}$ is an upper bound on the transmission delay. In the typical case where K(ϵ,τ) is an increasing function of τ and modulo the possibility to choose $s_{+}$, Equation (Equation 12) reduces to only one inequality:(13)$$\log_{2}K\left(\epsilon ,\tau^{\mathrm{pr}}+\tau_{+}^{\mathrm{tr}}\right)\leq b\left(\tau^{\mathrm{pr}}\right).$$

Anyhow, the inequalities depend on both the system (via K(·,·)) and channel (via $b\left(\cdot \right),\tau_{+}^{\mathrm{tr}}$). This means that correct operation in fact requests a certain level of conformity between the system and the channel.

The conditions for correct operation will be fleshed out in the next section. We conclude the section with a comment on observability and a remark on how the observer proceeds when a loss occurs.

**Observation** **1.**
*The following statements are true:*
*(i)* *Let the proposed observer correctly operate for a given ϵ>0 and $s_{+}$. Then, for any trajectory satisfying Equation* (Equation 1)*, the desired exactness of observation ϵ is ensured with respect to the norm ${∥\cdot ∥}_{P}$;**(ii)* 
*Let a communication channel be given. Also, let any ϵ>0 small enough be coupled with some $s_{+}$ so that the proposed observer with these ϵ and $s_{+}$ operates correctly via the channel at hand. Then, the system ${\left\{\varphi^{t}\right\}}_{t\in T}$ is observable on the set $S_{0}$ via this communication channel.*



All observability conditions that will be established in this paper are nothing but implications of ii) in this observation. This means that these conditions ensure the correct operation of the proposed observer modulo’s proper and feasible choice of its parameters. In other words, whenever these conditions are satisfied, a reliable state estimate can be obtained by means of this observer.

**Remark** **1.***Suppose that messages may be lost when transmitting over the communication channel. If a loss does occur, the message $q_{k}$ which was last received is used in Equation* (Equation 11) *not only during the intended time interval (from $s_{i}$ to $s_{i+1}$), but also during the subsequent time intervals until the next successful transmission. Certainly, there is no guarantee that the estimation accuracy will be within the desired ϵ on these extra intervals. However, as soon as a new message arrives, this accuracy is restored due to the very design of the observer. This robustness against losses is achieved without any feedback in the communication channel (i.e., the coder is not notified when losses occur on the channel), unlike many competing schemes [31,32,33,34].*

This remark extends on the situation where the message is not lost, but corrupted so that an incorrect $q_{k}$ is occasionally used in Equation (Equation 11).

## 4. Criteria for Observability of the System

A problem with the conditions (Equation 12) and (Equation 13) is that they use the function K(·,·) from (e.4), for which there is a lack of constructive techniques to compute, or at least to assess it from its “parents”: the dynamics $\left\{\varphi^{t}\right\}$, and the set $S_{0}$. In this section, we make a first step to overcome this deficit; whereas the function K(·,·) is a by-product of the coalesce of the dynamics and set, we re-master the conditions into a form where separate characteristics of the dynamics and the set are employed.

### 4.1. The Size of Finite Covering

Inspired by (e.4), we start with the question: How many balls of a common radius δ are needed to cover a given bounded set? Though not articulated thus far, our interest in fact focuses on the high exactness of estimation δ≈0. This, in turn, motivates asymptotical analysis as δ→0. A response to these concerns is partly given by the concept of an upper box-counting dimension ${\bar{d}}_{B}$, which is defined as follows.

**Definition** **3**([21])**.**
*The upper box-counting dimension ${\bar{d}}_{B}\left(F\right)$ of a bounded set $F\subset R^{n}$ is given by*
(14)$$\begin{array}{c} {\bar{d}}_{B}\left(F\right):=\underset{\delta \to 0}{lim sup}\frac{\log N_{\delta }\left(F\right)}{-\log\delta }. \end{array}$$*Here, $N_{\delta }\left(F\right)$ can be defined in any of the following ways, with all of them resulting in a common value* (Equation 14)*:*
*(i)* The smallest number of closed balls of radius δ that cover F;*(ii)* The smallest number of closed balls of radius δ and centers in F that cover F;*(iii)* The smallest number of cubes of side δ that cover F;*(iv)* The number of δ-mesh cubes that intersect F;*(v)* The smallest number of sets of diameter at most δ that cover F;*(vi)* The largest number of disjoint balls of radius δ with centers in F.*Also, the quantity* (Equation 14) *does not depend on the choice of the norm in (i),(ii), (v), and (vi).*

It follows that for arbitrarily small ϰ>0, the number of δ-balls with centers in *F* that are needed to cover *F* does not exceed $\delta^{-({\bar{d}}_{B}\left(F\right)+\varkappa )}$ for all sufficiently small δ>0.

As is well-known [21], ${\bar{d}}_{B}\left(F\right)={\bar{d}}_{B}\left(\bar{F}\right)\leq n$ for any bounded set $F\subset R^{n}$ and ${\bar{d}}_{B}\left(F\right)=n$ if the interior of *F* is not empty, $F_{1}\subset F_{2}\Rightarrow {\bar{d}}_{B}\left(F_{1}\right)\leq {\bar{d}}_{B}\left(F_{2}\right)$, and ${\bar{d}}_{B}\left(F_{1}\cup \ldots \cup F_{k}\right)=\max\left\{{\bar{d}}_{B}\left(F_{1}\right),\ldots ,{\bar{d}}_{B}\left(F_{k}\right)\right\}$, $k\in Z^{+}$. The box-counting dimension may assume non-integer values; for example, ${\bar{d}}_{B}\left(F\right)=1/\log_{2}3$ for the middle-thirds of the Cantor set F⊂R.

Our particular interest is in dynamical systems and their invariant sets $S_{0}$ with ${\bar{d}}_{B}\left(S_{0}\right)<n$; this case does hold for some chaotic systems and complex attractors $S_{0}$.

### 4.2. Balance between the Initial and Forthcoming Estimation Exactness, Respectively

Now, we are going to study relations between the initial exactness δ of the state *x* estimate $\hat{x}$ and the implied forthcoming exactness ϵ during the time horizon of duration $s_{+}$. This study is aimed at building the component (e.3) of which the proposed observer is composed, among others. We recall that this component is a function $\delta (\epsilon ,s_{+})$ for which Equation (Equation 10) holds.

The *growth rate* of the system $\left\{\varphi^{t}\right\}$ on the set $S_{0}$ is defined to be:(15)$$g\left(S_{0}\right):=\lim_{\delta \to 0}\underset{t\to \infty }{\bar{\lim}}t^{-1}\log_{2}\underset{\theta \in T:\theta \leq t}{\sup}\underset{x\in B_{\delta }\left(y\right),y\in S_{0}}{\sup}∥A_{\theta }\left(x\right)∥,\;\mathrm{where}\;A_{\theta }\left(x\right):=\frac{\partial \varphi^{\theta }}{\partial x}\left(x\right)$$
is the Jacobian matrix of the map $\varphi^{\theta }\left(\cdot \right)$ at point *x* and $\log_{2}\infty :=\infty$. It is well-defined for all sufficiently small δ, since x∈S in Equation (Equation 15), thanks to the following.

**Lemma** **2.**
*There exists $\delta^{0}>0$ such that $B_{\delta^{0}}\left(y\right)\subset S$ for any $y\in {\bar{S}}_{0}$.*


The proof of this lemma is trivial and thus omitted from this document.

In Equation (Equation 15), the limit $\lim_{\delta \to 0}$ exists since the subsequent quantity decays as δ decreases. Since all norms ∥·∥ in the space of n×n-matrices are equivalent, it is easy to see that $g(S_{0})$ does not depend on the choice of the norm.

Among other components, the proposed observer uses a function $\delta (\epsilon ,s_{+})$ with a special property described in (e.3). Now, we show how such a function can be built from $g(S_{0})$.

**Lemma** **3.**
*Let $g(S_{0})<\infty$. For any $\hat{g}>g\left(S_{0}\right)$ and any positive definite n×n-matrix P, there exists a function $\delta (\epsilon ,s_{+})$ with the property (e.3) that is given by*
(16)$$\delta \left(\epsilon ,s_{+}\right)=\epsilon 2^{-\hat{g}s_{+}}$$
*for all sufficiently small ϵ>0 and sufficiently large $s_{+}$.*


The proof of this lemma is provided in Appendix A.

### 4.3. Correct Operation of the Observer and a Criterion for Observability

By bringing the pieces together, we arrive at the following.

**Proposition** **2.***Suppose that Assumptions 1–3 hold and the system has a finite growth rate $g(S_{0})$ on the set $S_{0}$. Consider a communication channel with capacity c. If*(17)$$c>g\left(S_{0}\right){\bar{d}}_{B}\left(S_{0}\right),$$*the system ${\left\{\varphi^{t}\right\}}_{t\in T}$ is observable on the set $S_{0}$ via this communication channel in the sense of Definition* 2.

The proof of this proposition is provided in Appendix A. The previous inequality strongly resembles other inequalities in the context of entropy in dynamical systems that link dimensions, Lyapunov exponents, and entropy (see [23,24,25]).

**Remark** **2.***The bounded transmission delay $\tau_{j}^{\mathrm{tr}}$ from Equation* (Equation 9) *and its upper bound from (c.5) do not affect the condition* (Equation 17) *for observability.*

## 5. Constructive Estimates and Analytical Bounds

In this section, we make the next and final step for obtaining tractable conditions for observability. The road to this is via the development of techniques for assessing growth rate and the box-counting dimension. A technique will be employed in both these cases that is similar in spirit to the second Lyapunov method.

### 5.1. Lyapunov-Like Function

The characteristic trait of the classic Lyapunov function v(·) is its decay along the trajectories of the system. In the current context, we are not interested in such a decay. Instead, our interest is focused on the rate at which an infinitesimally small ball is expanded under the transition mapping $\varphi^{t}$. The smallness implies that this mapping is well-approximated by the first two terms of its Taylor series, and so the rate in question is nothing but the expansion rate due to the Jacobian matrix $A_{t}\left(x\right)$ defined in Equation (Equation 15). The deformation of a ball under a linear mapping *A* is described by the singular values of *A*; in particular, the maximal of them is the norm of *A* and may be used in Equation (Equation 15). If *P* is a symmetric positive definite matrix and $R^{n}$ is endowed with the *P*-related norm ${∥\cdot ∥}_{P}$, these values are the square roots of the solutions of the algebraic equation $\det[A^{⊺}PA-P]=0$ repeated in accordance with their algebraic multiplicities and ordered from large to small. With these in mind, we introduce a function v(·):S→R with special properties whose description uses the *t*-step increment of this function:(18)$$\Delta^{t}v\left(x\right):=v\left(\varphi^{t}\left(x\right)\right)-v\left(x\right).$$

**Assumption** **4.**
*There exist d∈[0,n], a bounded function v:S→R, constant Λ≥0, and symmetric positive definite matrix $P\in R^{n\times n}$, such that*
(19)$$\begin{array}{cc} \Delta^{t}v\left(x\right)+\sum_{i=1}^{\left\lfloor d\right\rfloor }\log_{2}\lambda_{i}\left(t,x\right)+\left(d-\left\lfloor d\right\rfloor \right)\log_{2}\lambda_{\left\lfloor d\right\rfloor +1}\left(t,x\right)\leq \Lambda t, & \forall x\in S,\;\forall t\in T:1\geq t>0, \end{array}$$
*where $\lambda_{1}\left(t,x\right)\geq \cdots \geq \lambda_{n}\left(t,x\right)$ are the roots of the algebraic equation*
(20)$$\det\left[A_{t}{\left(x\right)}^{⊺}PA_{t}\left(x\right)-\lambda P\right]=0$$
*repeated in accordance with their algebraic multiplicities and ordered from large to small, and $\log_{2}0-\infty$.*


In the discrete-time case, only t=1 is concerned in Equation (Equation 19). In the continuous-time case, Equation (Equation 19) is imposed only within the finite time horizon of duration 1.

With $P=I_{n}$, Equation (Equation 20) reduces to $\det\left[A_{t}{\left(x\right)}^{⊺}A_{t}\left(x\right)-\lambda I_{n}\right]=0$ and $\lambda_{i}\left(x,t\right)$ are the squares of the standard singular values of the Jacobian matrix $A_{t}\left(x\right)$. For a generic *P*, the roots $\lambda_{i}\left(x,t\right)$ can also be reduced to standard singular values. Indeed, let *U* be the symmetric and positive definite “square root” of the symmetric and positive definite matrix $P=U^{2}$. The solutions of Equation (Equation 20) are evidently identical to those of $\det\left[U^{-1}A_{t}{\left(x\right)}^{⊺}UUA_{t}\left(x\right)U^{-1}-\lambda I_{n}\right]=0$, and so the $\lambda_{i}\left(x,t\right)$’s are the squares of the ordinary singular values of the matrix $UA_{t}\left(x\right)U^{-1}$. This matrix is similar to $A_{t}\left(x\right)$, and so these two matrices represent a common linear transformation in various bases. Thus, the role of *P* is, in fact, that of a linear coordinate transformation in pursuit of ease of building Λ and v(·).

Assumption 4 will be utilized for assessment of both quantities that we are interested in. Specifically, it will be used with d=1 and arbitrary Λ to upper-estimate the growth rate (Equation 15) of the system; this estimate is given by Λ. With Λ=0 and some d∈[0,n], it will be used to establish an upper bound on the upper box dimension of the invariant set $S_{0}$; this bound is given by *d*.

In the case of a continuous time system, the next proposition provides an alternative to computing the transition maps $\varphi^{t},t\in \left(0,1\right]$ and checking infinitely many inequalities (Equation 19), each for its own t∈(0,1], when verifying Assumption 4. To state this proposition, we introduce the Jacobian matrix of the right-hand side in Equation (Equation 2):$$\begin{array}{c} J\left(x\right)=\frac{\partial f}{\partial x}\left(x\right). \end{array}$$

**Proposition** **3.***Let there exist d∈[0,n], a continuously differentiable bounded function w:S→R, constant Γ≥0, and a symmetric positive definite matrix $P\in R^{n\times n}$, such that*(21)$$\begin{array}{cc} \dot{w}\left(x\right)+\sum_{i=1}^{\left\lfloor d\right\rfloor }\gamma_{i}\left(x\right)+\left(d-\left\lfloor d\right\rfloor \right)\gamma_{\left\lfloor d\right\rfloor +1}\left(x\right)\leq \Gamma , & \forall x\in S, \end{array}$$*where $\dot{w}\left(x\right)=\frac{\partial w}{\partial x}f\left(x\right)$ and $\gamma_{i}\left(x\right)$ are the solutions of the algebraic equation*(22)$$\begin{array}{c} \det\left(J{\left(x\right)}^{⊺}P+PJ\left(x\right)-\gamma P\right)=0 \end{array}$$*ordered from largest to smallest $\left(\gamma_{1}\left(x\right)\geq \cdots \geq \gamma_{n}\left(x\right)\right)$ and repeated in accordance with their algebraic multiplicity. Then, Assumption 4 holds with the particular P and d of Equation* (Equation 22)*, $v\left(x\right)=\frac{w(x)}{\ln2}$ and $\Lambda =\frac{\Gamma }{\ln2}$.*

This result is proved in Appendix B.

### 5.2. Analytical Upper Bound on the System’s Growth Rate and Related Conditions for Observability

**Proposition** **4.***Let Assumptions 1–4 hold with d=1 and Λ≥0 in the last of them. Then, the growth rate* (Equation 15) *of the system on $S_{0}$ obeys the following bound:*
$$g\left(S_{0}\right)\leq \frac{\Lambda }{2}.$$

The proof of this proposition is provided in Appendix B.

By combining Propositions 2 and 4, we arrive at the following.

**Theorem** **1.***Suppose that Assumptions 1–4 hold with d=1 and Λ≥0 in the last of them, and consider a communication channel with capacity c. If*(23)$$c>\frac{\Lambda {\bar{d}}_{B}\left(S_{0}\right)}{2},$$*the system ${\left\{\varphi^{t}\right\}}_{t\in T}$ is observable on the set $S_{0}$ via this communication channel in the sense of Definition* 2.

The observation schemes proposed in [39,40] can sometimes work under the channel rates smaller than that given in Theorem 1. This improved rate comes at a price: these schemes are not robust against losses in the communication channel.

In [7], the observer requires some feedback in the channel and a channel rate of the form $n\log_{2}L$, where *L* is the Lipschitz constant of the mapping $\varphi^{1}$. The estimate (Equation 23) is less conservative, both because L≥Λ/2 (if *L* is related to a norm of the form ${∥\cdot ∥}_{P}$) and $n\geq {\bar{d}}_{B}\left(S\right)$, with ≥↦> in some cases. Moreover, the scheme from [7] does not enjoy robustness against losses in the communication channel.

Finally, Corollary 6.2.1 of [29] provides an estimate for the topological entropy by using a result from [41], which is identical to our estimate of the rate *c* with identical assumptions.

### 5.3. Analytical Bounds on the Upper Box Dimension and Final Conditions for Observability

A drawback of Theorem 1 is that it uses the upper box dimension, whereas there are no general techniques to compute this dimension analytically. To compensate for this drawback, we will use results from [28,30] to replace the upper box dimension by its upper estimate in the form of another well-known kind of dimension, i.e., the so-called Lyapunov dimension. The benefit from this is that the latter can be estimated analytically.

We start by introducing the necessary definitions, including those of the Lyapunov dimension of a map in a point, of a map over a set, and of a dynamical system. Next, we will recall the required results from [28,30], and finally, we will provide the general results of this paper, which offers analytical conditions for observability under a finite communication bit-rate.

**Definition** **4.***For any t∈T, the* singular value function *of $A_{t}\left(x\right)$ of order d∈[0,n] at point $x\in R^{n}$ is defined as*
$$\begin{array}{c} \omega_{d}\left(A_{t}\left(x\right)\right):=\left\{\begin{array}{cc} 1, & d=0,\\ \sigma_{1}\left(A_{t}\left(x\right)\right)\cdots \sigma_{d}\left(A_{t}\left(x\right)\right), & d\in \{1,..,n\},\\ \sigma_{1}\left(A_{t}\left(x\right)\right)\cdots \sigma_{\left\lfloor d\right\rfloor +1}{\left(A_{t}\left(x\right)\right)}^{d-\left\lfloor d\right\rfloor }, & d\in (0,n)\setminus \{1,..,n-1\}. \end{array}\right. \end{array}$$
*Here $\sigma_{1}\left(A\right)\geq \ldots \geq \sigma_{n}\left(A\right)$ are the singular values of the n×n-matrix A.*


**Definition** **5**([30])**.**
*For any t∈T, the Lyapunov dimension of the map $\varphi^{t}\left(\cdot \right)$ at the point x∈S is given by*
$$\begin{array}{c} d_{L}\left(\varphi^{t},x\right):=\sup\left\{d\in \left[0,n\right]:\omega_{d}\left(A_{t}\left(x\right)\right)\geq 1\right\}. \end{array}$$

**Definition** **6**([30])**.**
*For any t∈T, the Lyapunov dimension of the map $\varphi^{t}\left(\cdot \right)$ with respect to the invariant set $S_{0}$ is given by*
$$\begin{array}{c} d_{L}\left(\varphi^{t},S_{0}\right):=\underset{x\in S_{0}}{\sup}d_{L}\left(\varphi^{t},x\right)=\underset{x\in S_{0}}{\sup}\sup\left\{d\in \left[0,n\right]:\omega_{d}\left(A_{t}\left(x\right)\right)\geq 1\right\}. \end{array}$$

**Definition** **7**([30])**.**
*The Lyapunov dimension of the dynamical system ${\left\{\varphi^{t}\right\}}_{t\geq 0}$ with respect to the invariant set $S_{0}$, is defined as*
$$\begin{array}{c} d_{L}\left({\left\{\varphi^{t}\right\}}_{t\in T},S_{0}\right):=\underset{t\in T}{\inf}\underset{x\in S_{0}}{\sup}d_{L}\left(\varphi^{t},x\right)=\underset{t\in T}{\inf}\underset{x\in S_{0}}{\sup}\sup\left\{d\in \left[0,n\right]:\omega_{d}\left(A_{t}\left(x\right)\right)\geq 1\right\}. \end{array}$$

For the sake of completeness, we provide the results that we borrowed from [28,30].

**Theorem** **2**([28])**.**
*Let Assumptions 1–3 hold. Then,*
$$\begin{array}{c} {\bar{d}}_{B}\left(S_{0}\right)\leq d_{L}\left(\varphi^{1},S_{0}\right). \end{array}$$

**Corollary** **1**([28])**.**
*Let the hypotheses of Theorem 2 be true. Then for all t∈T:t≥1,*
$$\begin{array}{c} {\bar{d}}_{B}\left(S_{0}\right)\leq d_{L}\left(\varphi^{t},S_{0}\right). \end{array}$$

The following proposition is essentially a reformulation of Theorem 2 from [30].

**Proposition** **5.**
*Let Assumptions 1–3 be true. Suppose also that Assumption 4 holds with some d∈[0,n] and Λ=0. Then, for sufficiently large l>0, the following inequality is valid:*
(24)$$d_{L}\left({\left\{\varphi^{t}\right\}}_{t\in T},S_{0}\right)\leq d_{L}\left(\varphi^{l},S_{0}\right)\leq d.$$


The proof of this proposition is provided in Appendix B.

In some cases, the inequalities in Equation (Equation 24) take place as equalities. Specifically, the following proposition is valid, which is a reformulation of Proposition 3 and Corollary 3 from [30].

**Proposition** **6**([30])**.**
*Suppose that at one of the equilibrium points of the dynamical system ${\left\{\varphi^{t}\right\}}_{t\in T}:x_{\mathrm{eq}}\equiv \varphi^{t}\left(x_{\mathrm{eq}}\right)$, the matrix $A_{1}\left(x_{\mathrm{eq}}\right)$ has the simple real eigenvalues $\lambda_{1}\left(x_{\mathrm{eq}}\right),\cdots ,\lambda_{n}\left(x_{\mathrm{eq}}\right)$. Let us consider a non-singular matrix U, such that*
(25)$$\begin{array}{c} UA\left(x_{\mathrm{eq}}\right)U^{-1}=\mathrm{diag}\left(\lambda_{1}\left(x_{\mathrm{eq}}\right),\cdots ,\lambda_{n}\left(x_{\mathrm{eq}}\right)\right) \end{array}$$
*where $|\lambda_{1}\left(x_{\mathrm{eq}}\right)\left|\geq \cdots \geq \right|\lambda_{n}\left(x_{\mathrm{eq}}\right)|$, which matrix does exist thanks to the first assumption of the proposition. Let $\varphi_{U}:w\to U\varphi^{1}\left(U^{-1}w\right)$ be the transition mapping after the linear coordinate change. Suppose that Assumption 4 holds with some d and Λ=0, and additionally, we have*
$$\begin{array}{c} d_{L}\left(\varphi_{U},Ux_{\mathrm{eq}}\right)=d. \end{array}$$
*Then, for any compact invariant set $S_{0}∋x_{\mathrm{eq}}$ of ${\left\{\varphi^{t}\right\}}_{t\in T}$, the following equation holds*
$$\begin{array}{c} d_{L}\left({\left\{\varphi^{t}\right\}}_{t\in T},S_{0}\right)=d. \end{array}$$


Now we are in a position to state the main result of the paper, which is clear from Theorem 1 and Proposition 5.

**Theorem** **3.***Let Assumptions 1–3 be true. Suppose also that Assumption 4 holds twice: first, with d=1 and some $\Lambda =\bar{\Lambda }\geq 0$ and second, with Λ=0 and some $d=\bar{d}\in \left[0,n\right]$. Consider a communication channel with capacity c. If*(26)$$c>\frac{\bar{\Lambda }\bar{d}}{2},$$*the system ${\left\{\varphi^{t}\right\}}_{t\in T}$ is observable on the set $S_{0}$ via this communication channel in the sense of Definition* 2.

## 6. Examples

In this section, we apply the previous theory to two celebrated prototypical chaotic systems: the smoothened Lozi map and the Lorenz system. For the smoothened Lozi map, we will compute the Lyapunov dimension and provide a bound on the channel rate above where the associated dynamical system is observable via the channel at hand. We will then test this bound via computer simulations of the proposed observer to show that the established theoretical rates are close to the actual practical rates. For the Lorenz system, we borrow upper estimates of the Lyapunov dimension and the largest singular value of the Jacobian from [39,42], respectively, to provide a bound on the channel rate by using Theorem 3. Like in the previous example, we will also test this bound via computer simulations.

### 6.1. The Smoothened Lozi Map

The Lozi map [43,44] is a modification of the Henon map. The Lozi map is not continuously differentiable, and so does not meet Assumption 3. We examine its continuously differentiable analog introduced in [45] by smoothing the Lozi map at the fracture point. The respective smoothened map acts according to the following formula
(27)$$\begin{array}{c} \varphi_{\alpha }:\left(\begin{array}{c} x_{1}\\ x_{2} \end{array}\right)\to \left(\begin{array}{c} 1-af_{\alpha }\left(x_{1}\right)+bx_{2}\\ x_{1} \end{array}\right), \end{array}$$
where *a*, *b*, and α≪1 are positive parameters and
(28)$$\begin{array}{cc} f_{\alpha }\left(x\right) & =\left\{\begin{array}{cc} |x|, & \mathrm{if}\;|x|\geq \alpha ;\\ \frac{x^{2}}{2\alpha }+\frac{\alpha }{2}, & \mathrm{if}\;|x|<\alpha . \end{array}\right. \end{array}$$

If 1+a−b>0 and $\alpha <{\left(a+1-b\right)}^{-1}$, the smoothened Lozi map has an equilibrium
$$\begin{array}{c} x_{+}=\left(\frac{1}{1+a-b},\frac{1}{1+a-b}\right). \end{array}$$

If 1−a−b<0 and $\alpha <{\left(a-b-1\right)}^{-1}$ in addition to the previous inequalities, there exists a second equilibrium
$$\begin{array}{c} x_{-}=\left(\frac{1}{1-a-b},\frac{1}{1-a-b}\right). \end{array}$$

In this section, we adopt the following.

**Assumption** **5.**
*The following inequalities hold:*
$$\begin{array}{c} a,b,\alpha >0,\\ 1-a<b<1,\\ \alpha <{\left(a+1-b\right)}^{-1}. \end{array}$$


This assumption implies that two equilibria exist, and that they are unstable. Moreover, b<1 ensures $d_{L}\left({\left\{\varphi_{\alpha }^{t}\right\}}_{t\geq 0},K\right)<2$ for the associated discrete-time dynamical system. We start by giving more insight into the Lyapunov dimension of the smoothened Lozi map.

**Theorem** **4.***Let Assumption 5 hold. Then, for any compact invariant set $S_{0}$ of the map* (Equation 27)*, the following inequality is valid*
(29)$$\begin{array}{c} d_{L}\left({\left\{\varphi_{\alpha }^{t}\right\}}_{t\in T},S_{0}\right)\leq \bar{d}:=2-\frac{\log_{2}b}{\log_{2}\left(\sqrt{a^{2}+4b}-a\right)-1} \end{array}$$
*and Assumption 4 holds with Λ=0 and $d=\bar{d}$. Moreover, if $x_{+}\in S_{0}$, inequality* (Equation 29) *holds as equality:*
$$\begin{array}{c} d_{L}\left({\left\{\varphi_{\alpha }^{t}\right\}}_{t\in T},S_{0}\right)=2-\frac{\log_{2}b}{\log_{2}\left(\sqrt{a^{2}+4b}-a\right)-1}. \end{array}$$

The proof of this theorem is provided in Appendix C.

In order to use the observer from Section 3 for the smoothened Lozi map (Equation 27), we need to choose a compact and invariant set $S_{0}$. For the original (i.e., non-smooth) Lozi map, such a set exists whenever the following conditions are met: [46]
(30)$$\begin{array}{c} 0<b<1, \end{array}$$
(31)$$\begin{array}{c} a>0, \end{array}$$
(32)$$\begin{array}{c} 2a+b<4, \end{array}$$
(33)$$\begin{array}{c} b<\frac{a^{2}-1}{2a+1}, \end{array}$$
(34)$$\begin{array}{c} a\sqrt{2}>b+2. \end{array}$$

Moreover, when the previous inequalities hold, the set $S_{0}$ is the closure of the unstable manifold of the unstable equilibrium $x_{+}$. It is still unknown whether they guarantee the same for the smoothened Lozi map (Equation 27). To the best of the authors’ knowledge, no conditions that guarantee the existence of such a set for the map (Equation 27) are available in the literature. In the following, we will assume that Equations (Equation 30)–(Equation 34) are sufficient to ensure the existence of a compact and invariant set $S_{0}$. Our simulations with the parameters a=1.7 and b=0.3, $\alpha =10^{-5}$, which verify Equations (Equation 30)–(Equation 34), provide evidence, illustrated in Figure 2, in favor of this hypothesis.

By combining Theorem 3 with Theorem 4 and an estimate of the largest singular value of the concerned Jacobian given in [40], we arrive at the following.

**Corollary** **2.***Let Assumption 5 hold, and let $S_{0}$ be a compact invariant set of the smoothened Lozi map* (Equation 27)*. Then, the associated dynamical system is observable on the set $S_{0}$ via any communication channel whose capacity*
$$c>\frac{\left[2\log_{2}\left(\frac{\sqrt{a^{2}+4b}-a}{2}\right)-\log_{2}b\right]\left[\log_{2}\left(\frac{\sqrt{a^{2}+4b}+a}{2}\right)\right]}{\log_{2}\left(\frac{\sqrt{a^{2}+4b}-a}{2}\right)}.$$

**Proof.** Theorem 13 from [18] yields that Assumption 4 holds with d=1 and
$$\begin{array}{c} \Lambda =2\log_{2}\left(\sqrt{a^{2}+4b}+a\right)-2. \end{array}$$Theorems 3 and 4 complete the proof. □

Corollary 2 implies that for these parameters, the associated dynamical system is observable for any channel rate above 1.2013 bits/s. To verify whether this lower bound on the channel rate can be improved, we employed the observer from Section 3 whose parameters were experimentally tuned to ensure a pre-specified exactness of observation ϵ during the first 1000 steps. The following values were considered: ϵ=0.5,0.2,0.1,0.05. An accompanying objective of experimentally tuning was to minimize the size of the alphabet *K* employed for data encoding, or, in other words, the channel capacity $c^{*}$ requested by the observer. The best values of the capacity can be found in Table 1. It shows that for high exactness, the system can be observed with a channel rate slightly below the theoretical estimate. For the lowest exactness, the experimental rate barely exceeds the theoretical bound. However, in any case, this bound seems to be pretty close to the experimental result.

### 6.2. The Lorenz System

In this section, we apply our previous theoretical results to the Lorenz system. The Lorenz system [47] is a well-known example of a continuous-time system where, for certain values of its parameters, it displays chaotic behavior. The system equations are:(35)$$\begin{array}{c} \left\{\begin{array}{ccc} {\dot{x}}_{1} & = & -\sigma x_{1}+\sigma x_{2},\\ {\dot{x}}_{2} & = & \rho x_{1}-x_{1}x_{3}-x_{2},\\ {\dot{x}}_{3} & = & x_{1}x_{2}-\beta x_{3}, \end{array}\right. \end{array}$$
where σ,ρ, and β are positive parameters. If ρ<1, the system has a single globally asymptotically stable equilibrium: the origin. For ρ>1, this equilibrium becomes a hyperbolically unstable saddle-point. In addition, two equilibria appear. In this paper, we assume ρ>1. We will apply our findings to the system with a chaotic attractor as an invariant set. As is well-known [48], the conditions on the parameters (σ>0,β>0,ρ>1) suffice to ensure the presence of a compact invariant set. Moreover, to compute the Lyapunov dimension, we adopt the following assumption which is taken from [42].

**Assumption** **6.**
*Let the following hold:*
$$\begin{array}{cc} \rho & >1,\\ \rho & \geq \frac{\beta^{3}-2\beta^{2}+6\beta^{2}\sigma -3\beta \sigma^{2}-6\beta \sigma +\beta }{3\sigma^{2}}+1,\\ \sigma \rho & >(\beta +1)(\beta +\sigma ), \end{array}$$
*and either*
$$\begin{array}{c} \sigma^{2}\left(\rho -1\right)\left(\beta -4\right)\leq 4\sigma \left(\sigma \beta +\beta -\beta^{2}\right)-\beta {\left(\beta +\sigma -1\right)}^{2}, \end{array}$$
*or the following equation has two distinct solutions, ν*
(36)$$\begin{array}{c} {\left(2\sigma -\beta +\nu \right)}^{2}\left(\beta {\left(\beta +\sigma -1\right)}^{2}-4\sigma \left(\sigma \beta +\beta -\beta^{2}\right)+\sigma^{2}\left(\rho -1\right)\left(\beta -4\right)\right)+4\beta \nu \left(\sigma +1\right)(\beta {\left(\beta +\sigma -1\right)}^{2}\\ -4\sigma \left(\sigma \beta +\beta -\beta^{2}\right)-3\sigma^{2}\left(\rho -1\right))=0 \end{array}$$
*and*
$$\begin{array}{c} \left\{\begin{array}{c} \sigma^{2}\left(\rho -1\right)\left(\beta -4\right)>4\sigma \left(\sigma \beta +\beta -\beta^{2}\right)-\beta {\left(\beta +\sigma -1\right)}^{2}\\ \nu_{1}>0 \end{array}\right., \end{array}$$
*where $\nu_{1}$ is the largest root of Equation (Equation 36).*


Any solution of the Lorenz system that starts at t=0 can be extended on [0,∞) [29], and thus has the extendability property discussed just after Equation (Equation 2). Hence, the differential Equations (Equation 35) give rise to a dynamical system on $S:=R^{3}$ in the sense of Section 2.1.

**Proposition** **7.**
*Let Assumption 6 hold, and let $S_{0}$ be a compact invariant set of the Lorenz system. Then, this system is observable on the set $S_{0}$ via any communication channel whose capacity is:*
$$c>\frac{\left[\sqrt{{\left(\sigma -1\right)}^{2}+4\rho \sigma }-\sigma -1\right]\left[3\sqrt{{\left(\sigma -1\right)}^{2}+4\rho \sigma }-2\beta +\sigma +1\right]}{2\ln2\left[\sqrt{{\left(\sigma -1\right)}^{2}+4\rho \sigma }+\sigma +1\right]}.$$


**Proof.** Since the right-hand side of the equations in Equation (Equation 35) are polynomial, Assumptions 1 and 3 hold by Proposition 1; Assumption 2 is true due to the choice of $S_{0}$. It is easy to see by inspection of the proof of Theorem 4.3 from [39] that the assumptions of Proposition 3 hold with d=1,
$$\begin{array}{c} \Gamma =\sqrt{{\left(\sigma -1\right)}^{2}+4\rho \sigma }-\sigma -1, \end{array}$$
and the matrix *P* defined by (16) in [39]. So Proposition 3 guarantees that Assumption 4 holds with d=1 and $\Lambda =\frac{1}{2\ln2}\left[\sqrt{{\left(\sigma -1\right)}^{2}+4\rho \sigma }-\sigma -1\right]$. As is shown in Section 4 from [42], Assumption 4 also holds with Λ=0 and
$$\begin{array}{c} d=\bar{d}=3-\frac{2(\sigma +\beta +1)}{\sqrt{{\left(\sigma -1\right)}^{2}+4\sigma \rho }+\sigma +1}. \end{array}$$Theorem 3 completes the proof. □

We have performed simulation studies similar to those carried out for the previous example. Starting from various initial conditions in $S_{0}$, we have simulated the observer for various chosen ϵ, each with its own chosen δ and associated covering. In our simulations, we used σ=10, ρ=28, $\beta =\frac{8}{3}$, which verify Assumption 6. For these parameters, the theoretical rate bound is c>40.975 bit/s. The results of the simulations can be seen in Table 2. Once again, it can be seen that the experimentally found rate is below or very close to the theoretical rate. This confirms that our theoretical results correctly predict the rate.

## 7. Conclusions

In this paper, we have presented an observer for both discrete and continuous-time nonlinear systems. We have provided bounds on the necessary data-rates to implement the observer. We have proven that this observer can be implemented on any channel with a finite delay parameter and a channel rate $c>\Lambda {\bar{d}}_{B}\left(S_{0}\right)/2$, where Λ/2 is an upper bound on the largest singular value of the Jacobian and ${\bar{d}}_{B}\left(S_{0}\right)$, the upper box dimension of the compact invariant set of the system. By combining results from several other papers, we have provided an analytical bound to the channel rate that depends on the Lyapunov dimension, rather than the upper box dimension. These analytical bounds have been computed for the smoothened Lozi map and the Lorenz system. For the smoothened Lozi map, we computed the Lyapunov dimension. Simulations of the observer on both of these systems have proven that the theoretical rate is closely related to the actual rate required to implement the observer.

## Figures and Tables

**Figure 1 entropy-21-00282-f001:**
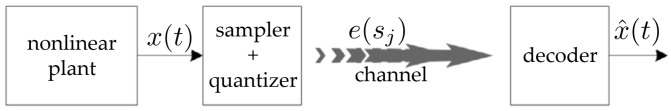
Structure of the observer.

**Figure 2 entropy-21-00282-f002:**
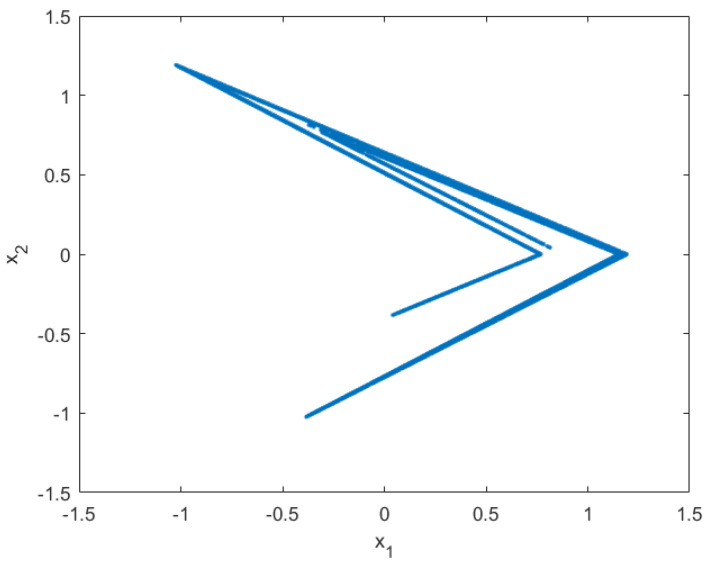
Typical trajectory with 10,000 steps of the smoothened Lozi map with a=1.7, b=0.3, and $b=10^{-5}$.

**Table 1 entropy-21-00282-t001:** Results of the simulations on the smoothened Lozi map.

	ϵ=0.2	ϵ=0.1	ϵ=0.075	ϵ=0.05
*K* (1)	$1\times 10^{6}$	$2\times 10^{6}$	$2\times 10^{7}$	$3.5\times 10^{7}$
$c^{*}$ (bits/s)	1.0924	1.1499	1.169431	1.212770

**Table 2 entropy-21-00282-t002:** Results of the simulations on the Lorenz system.

	ϵ=0.2	ϵ=0.1	ϵ=0.075	ϵ=0.05
*K* (1)	364,758	714,701	1,448,880	$3.5\times 10^{7}$
$c^{*}$ (bits/s)	19.814	30.739	34.614	43.714

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
