# Peer review of "Data-Rate Constrained Observers of Nonlinear Systems"

_entropy, 2019, doi:10.3390/e21030282_

Round 1

Reviewer 1 Report

See the attached PDF.

Author Response

Dear Reviewer,

Please find the answers to the many useful and relevant comments and suggestions you made in the attached file.

We thank you for your time and consideration.

Sincerely,

Quentin Voortman, Alexander Pogromsky, Alexey Matveev, and Henk Nijmeijer

Reviewer 2 Report

This is an extremely interesting paper to me to provide observability conditions in terms of an observer and a channel, relating classical dynamical systems concepts such as box dimension but more so Lyapunov dimension here, to channel capacity for describing observability.  The end results are striking and satisfying that they come all the way to state explicit bounds for the channel capacity based in terms of parameters of the dynamical system (corollary 2 and proposition 7 pages, 15-16).

There are several places I became frustrated in notation and so this causes me to become disengaged from the details in a way that makes me uncomfortable to make a detailed assessment.  In particular, the early definitions of Eqs 3-8 are so central to what follows. 

-what is A_j? A is the alphabet A={0,1,...K} but A_j is stated as the alphabet at e(s_j) at time s_j?  This is not making sense to me.  You are changing alphabet at each time? Or are we talking about an n-cylinder of a symbolic sequence.  This point alone may be the difficulty in detailed reading for me.

-What is the {\cal Q} symbol in Eq 5 denoting as the function modifying the stated set?

-Same for the {\cal D}, and same with {\cal S} in Eq. 3.

-I am not following how the function b would be computed.

-T\in reals is a time interval (rather than a single time instance a point?)

-In page 7, the statement "the alphabet substantiates the numbering of the balls" - again my question is the alphabet changing in time?  I am thinking in terms of symbol dynamics where an alphabet is fixed and the symbols from the alphabet change in time but that picture is not consistent with what I am reading.

-later on page 7 it does say "fixed alphabet {1,...,K}

-uneven presentation of details - sometimes more detail than needed (for me) like about what is the box dimension of the middle thirds cantor set (top of page 9) but not enough detail regarding what are central terms to the paper, like for example channel capacity, which I take to be as usual as we read say from Thomas-Cover on Shannon information theory.

-Its from the results I start to catch up to reverse engineer what must have been the earlier definitions and this is telling me the definitions are heavy on notation perhaps more than needed?  For example, Proposition 2, Eq 17 was very helpful to me to begin to catch up.

Mostly I have a strong feeling my only argument is with presentation style early in the paper that has made the very interesting results harder to follow in the most detailed way that this paper deserves.  I challenge the authors to clearly state in these topological entropy, information theoretic terms the central phrases "observable" and "channel" and a few others.

The results are very interesting.  As are some of the methods, for example the way the Lyapunov "direct" function method concept is adapted to this problem.  The final results with their analytic statements and then backchecking with numerics is impressive.

Also there are several awkward statements for their English grammar, such as "channel at hands" (page 13), "dynamical systems at hands" (assumption 1 page 3) and several other places but I am afraid I did not make a complete list.

I would recommend expanding the narration on the parts which were heavy on notation as I described as very interesting results as these deserve to be widely understood and read.  I anticipate a positive recommendation at that time.

Also I would point a paper that has some interesting relationship, Stajanovski et al., 1997 in the field of synchronization of oscillators is that in synchronizing two oscillators, the channel capacity of the coupling must exceed the Komolgorov-Sinai entropy of the driving system, Applications of symbolic dynamics in chaos synchronization

T Stojanovski, L Kocarev, R Harris

IEEE Transactions on Circuits and Systems I: Fundamental Theory and â€¦

1997

Author Response

(The authors gave the same response as above.)
